# COVID-19 Pandemic Influence on Healthcare Professionals

**DOI:** 10.3390/jcm10061280

**Published:** 2021-03-19

**Authors:** Anna Klimkiewicz, Adrianna Schmalenberg, Jakub Klimkiewicz, Agata Jasińska, Joanna Jasionowska, Weronika Machura, Marcin Wojnar

**Affiliations:** 1Department of Psychiatry, Medical University of Warsaw, Nowowiejska Str. 27, 00-665 Warsaw, Poland; anna.klimkiewicz@wum.edu.pl (A.K.); marcin.wojnar@wum.edu.pl (M.W.); 2Psychomedic Clinic, Jastrzębowskiego Str. 24, 02-783 Warsaw, Poland; adriannaschm@gmail.com; 3SWPS—University of Social Sciences and Humanities, Chodakowska Str. 19/31, 03-815 Warsaw, Poland; 4Department of Anesthesiology and Intensive Care, Military Institute of Medicine, Szaserów Str. 128, 04-141 Warsaw, Poland; wmachura@wim.mil.pl; 5Department of Science and Technology Transfer, Medical University of Warsaw, Żwirki i Wigury Str. 61, 02-091 Warsaw, Poland; agata.jasinska@wum.edu.pl; 6Department of Psychiatry, Nowowiejski Psychiatric Hospital, Nowowiejska Str. 27, 00-665 Warsaw, Poland; joanna.j@neostrada.pl; 7Addiction Center, Department of Psychiatry, University of Michigan, Rachel Upjohn Building, 4250 Plymouth Rd., Ann Arbor, MI 48109, USA

**Keywords:** COVID-19, healthcare professionals, depression, sleep disorders, alcohol consumption

## Abstract

During the pandemic, many healthcare professionals (HCPs) are overburdened by work and stress. The aim of the study was to examine alcohol intake, sleep disorders, and depressive symptoms of HCPs during the pandemic in comparison with the pre-pandemic period. Another goal was to indicate risk factors for mental state deterioration and an increase in alcohol use. A cross-sectional survey study was conducted from 1 April to 15 May 2020. HCPs (*n* = 158) completed questionnaires that probed for symptoms during and prior to the pandemic, including the Beck depression inventory (BDI), Social Support Scale (MOS-SSS), Athens insomnia scale (AIS), and Alcohol Timeline Followback (TLFB) calendar of alcohol consumption. Gender, age, education, marital status, work situation, income, participants’ and relatives’ COVID-19 diagnosis as correlates were analyzed. Depressive symptoms and insomnia became more severe during the pandemic among HCPs, while social support increased. The increase in depressive symptoms was even higher among women (OR 2.78, 95% CI 1.05–7.36; *p* = 0.04) and was also positively correlated with work reduction (*p* = 0.02); the presence of sleep disorders was correlated with female gender. Alcohol consumption increased during the pandemic, and was correlated with both more time spent at work and income increase. HCPs involved in the treatment of COVID-19 need support and attention due to the excessive stress load during pandemics, resulting in depression, insomnia, and increased alcohol intake.

## 1. Introduction

While the main focus is currently on patients with COVID-19, healthcare workers struggle with enormous stress and fear, as well as with stigma. Frontline healthcare personnel are at the highest risk of mental health problems. The studies conducted so far in the field of healthcare workers’ mental conditions in the face of the epidemic have indicated that a significant percentage of medical staff experience anxiety, depression, and trauma [1,2,3]. Cabakarpa et al., in a systematic review, concluded that the negative mental consequences of being at the frontline, fighting the pandemic, necessitate more attention, education, and the involvement of mental health professionals [4]. The stress of healthcare professionals (HCPs) in the COVID-19 pandemic has many causes, such as fear of the unknown, SARS-CoV-2 being highly contagious, uncertainty, stigma, organizational factors, etc. [5,6]. Such factors as depletion of personal protection equipment are other causes of anxiety and fear. Concerns will arise about the psychological adaptation and mental health of medical staff. For instance, healthcare workers in China, who were directly engaged in the diagnosis, treatment, and care of patients with COVID-19, were at a higher risk of insomnia, distress, as well as depressive and anxiety symptoms of depression [2]. It is already known that chronic stress, insomnia, and depression are strongly linked. For example, stress influences the metabolic, cardiovascular and immune systems. Those are further involved in the short- and long-term consequences of stress.

In the time of an epidemic, HCPs experience many stressors that affect well-being. The first group of stressors are those directly related to the nature of their job. The second group of stressors are those that apply to every human being in the general population. The fear of being infected, uncertainty of the future, fear of the deterioration of the financial situation, as well as illness and loss of loved ones, concern both HCPs and the general population. Both kinds of stressful stimulus have an essential impact on HCPs’ functioning and their mental condition. Importantly, the broader the knowledge is about SARS-CoV-2, COVID-19, and the entailed complications, the greater the fear for their own and their loved ones’ health in this group.

In a systematic review on the mental health of healthcare personnel during COVID-19 in Asia, the authors mentioned depression, insomnia and anxiety as the most common psychiatric effects of the COVID-19 pandemic [7]. There are some important reports originating from European countries as well, e.g., Italy [8], Spain [9], Norway [10] and Switzerland [11].

According to previous research, there are several important risk factors for HCPs’ mental health deterioration during a pandemic. Those that have been identified so far include female gender [12,13,14], and direct contact with the infected patients [15,16], especially with a lack of the appropriate personal protective equipment [17]. In addition, shorter work experience predicted a higher prevalence of anxiety and depressive symptoms [18]. Healthcare professionals that were either significantly younger or older than average were also at risk of mental disorders due to the pandemic [14,19,20].

Some researchers also indicated protective factors against the pandemic-induced worsening of mental state. The first of these is mental resilience [21], and the second is social and family support [22,23].

The linkage between the prevalence of chronic stress and depression has already been studied and confirmed, and many mechanisms connecting chronic stress and anxiety and depression have already been discovered and described [24,25,26]. Consequently, it can be expected that the increasing frequency of common mental health disorders (especially depression and anxiety) due to the pandemic will possibly be followed by increased alcohol consumption and alcohol use disorders. Chronic stress, such as that caused by the COVID-19 pandemic, can be considered an important risk factor of mental disorders for both genders. However, this phenomenon is particularly prevalent among women under both pandemic [12,13,14] and “non-pandemic” stress [27]. The coexistence of alcohol abuse and dependence with some mental disorders, especially mood and anxiety, has already been studied and fully described [28]. Higher rates of both anxiety and depression have been proven among alcohol abusers [29]. This link seems to be particularly important in the context of chronic stress in the face of a pandemic. There are also significant differences in the incidence rates of depressive disorders and alcohol dependence based on gender. This should be taken into consideration when analyzing the problem of their co-existence. Among women who are observed to experience anxiety or affective disorders, the risk of developing alcohol use disorders is greater, and alcohol dependence develops in a shorter period of time than among those without depressive or anxiety disorders [27,28,29]. Chronic stress is an important contributor to depression, insomnia, and alcohol abuse. As it occurs in people working in healthcare, it may, as a consequence, be a threat to both them and to their patients.

## 2. Objectives

While SARS-CoV-2 was spreading in Poland, the influence of the situation on the mental state of healthcare professionals was explored. The study aims to research whether insomnia, alcohol intake and symptoms of depression were influenced by the COVID-19 epidemic among healthcare professionals (HCPs). Moreover, the study aims at identifying factors related to alterations in mental state. It was examined whether changes in the abovementioned factors (insomnia, alcohol intake, and symptoms of depression) were associated with the following: gender, age, education, living alone vs. with someone, employment status, income, individuals’ and their relatives’ COVID-19 diagnosis, etc. To achieve those goals, current depressive and insomnia symptoms, as well as alcohol intake during the COVID-19 pandemic, were examined and compared with depressive symptoms, insomnia, and alcohol consumption reported prior to the start of the pandemic. Pre-pandemic and post-pandemic outbreak assessments have been conducted simultaneously. The longitudinal evaluation of mental health symptoms was not performed, as it was impossible due to the unexpected nature of the COVID-19 epidemic. It is hypothesized that there will be a significant change in the mental state and alcohol consumption of studied individuals as the effects of chronic stress unfold. Based on previous research we have indicated possible factors linked with mental state alteration, and further examined those factors.

## 3. Methods

### 3.1. Participants

The study was conducted online from 1st April until 15th May. The healthcare professionals (HCPs) group consisted of 158 participants, recruited among those medical professionals who worked with patients infected with SARS-CoV-2 during the study’s course. The HCPs group consisted of medical doctors, paramedics and nurses, who were recruited using the snowball method. In total, 186 HCPs were invited to participate. The refusal rate was as high as 15%. At the beginning, 20 HCPs from one of the Warsaw COVID-19 units were asked to participate. Those who took part in the study provided referrals to recruit another HCP working on a COVID-19 unit. Every HCP willing to participate and who met criteria of working in a pandemic frontline setting met the eligibility criteria, and was included in the analysis.

### 3.2. Procedure

The data were analyzed anonymously, and the study protocol was prepared in line with the recommendations of the Helsinki Declaration and has been reported for the approval of the Bioethics Committee at the Medical University of Warsaw. Participation consent was given by checking a box, whereby participants declared that they understand all the information and agree to the participation. Returning a questionnaire without the above-mentioned box checked was not accepted, thus the data contained in such a survey were not analyzed, and the entirety of the personal information in this form was permanently deleted from the database. In this way, it was guaranteed that only individuals who provided informed consent to the study participated in the project.

First, participants were asked to complete questionnaires about the time period prior to the pandemic (i.e., during the last two weeks of February 2020) to examine their mental condition prior to the spread of SARS-CoV-2 infections and the World Health Organization WHO’s announcement of the pandemic. Second, participants were asked the same questions regarding their current mental state (i.e., past two weeks). Questions were asked one after another to allow for a one-to-one comparison of symptoms over time. Each question was therefore answered twice (i.e., “now in the pandemic” and “then—before it started”). The pre-pandemic assessment time (i.e., the last two weeks of February 2020) was established arbitrarily. Mental state prior to the pandemic was not tested directly in real-time, given that the rapid spread of COVID-19 was unexpected. Thus, the subjects were asked to assess their mental state during February as accurately as they remembered it. None of the study participants reported difficulties with recalling that time period. Respondents were also asked whether they, or their relatives, were infected with SARS-CoV-2. Basic demographic data, as well as the financial and employment status, were collected. Participants were also asked whether themselves or any of the relatives were infected with SARS-CoV-2, as we found this to be an important factor altering the subject’s mental condition.

Alcohol consumption was evaluated using a follow-back calendar. Participants filled checkboxes, each corresponding to a two-week period starting from the pre-pandemic period of the last two weeks of February. For each of the total of 5 periods, individuals were asked to estimate their average daily alcohol consumption, counted in standard portions of 10 g of ethanol. The ranges were as follows: 0 portion, 1–2.5 portions, 2.6–7.5 portions, 7.6–10 portions, 10–20 portions, over 20 portions. The number of standard portions contained in most popular alcoholic beverages was presented as a simple graphic in the survey.

### 3.3. Measures

The questionnaire included standardized scales:The Beck depression inventory (BDI-II) was administered to assess depressive symptoms. The BDI-II consists of 21 self-report questions. There are four response variants that correspond to the increased intensity of the symptoms and which are scored on a 0 to 3 scale [30]. Higher scores correspond to greater depressive symptoms. The cut-off scores for the BDI-II are as follows: minimal range = 0–13 points; mild depression = 14–19 points; moderate depression = 20–28 points; severe depression = 29–63 points. The Polish adaptation of the scale (Cronbachs’ alpha = 0.91) by Łojek and Stańczak was used [31].The Athens insomnia scale (AIS) was administered to assess sleep disorders [32]. Higher scores indicate greater insomnia symptoms; 0–5 points is considered to be in the normal range, whereas scores of 6 points or higher indicate sleep disorders. The Polish adaptation of the scale (Cronbach’s alpha = 0.9) was used [33].The social support scale (MOS-SSS) was administered to assess social support [34]. Higher scores indicate greater social support for respondents. The scoring was as follows: 19–38 points = low score; 38–76 points = medium score; 76–95 points = high score. The Cronbach’s alpha for the scale was as high as 0.97.

The Cronbach’s alpha for all measures was higher than 0.7; therefore, this study can be considered reliable.

### 3.4. Statistical Analysis

Statistical analysis was conducted using the IBM SPSS Statistics 25 Package (IBM, Armonk, NY, USA). Non-parametric tests were used due to the non-normal distribution of the variables and because the groups had different numbers of participants. At the beginning, insomnia, depression and alcohol consumption were compared between the two analyzed time periods. Further, we assessed the correlation of factors, such as gender, age, education, living alone vs. with someone, employment status, income, and individuals’ and their relatives’ COVID-19 diagnosis, with insomnia, depression and alcohol consumption increase. The Wilcoxon test was used to check whether there are statistically significant differences between the two time periods for the analyzed variables. The analysis of the Spearman correlation allowed us to check the presence of a statistically significant relationship between the studied variables. Then, a multinomial logistic regression was performed for factors significantly correlated with the increase in depression and insomnia symptoms, and higher alcohol consumption. Significant outcomes from the logistic regression are presented in the results section.

## 4. Results

Demographic data are presented in Table 1.

The pandemic period resulted in the exacerbation of depressive and insomnia symptoms in the analyzed group of healthcare professionals, although the level of social support increased during the pandemic. All the mentioned differences were statistically significant. When assessing insomnia using the Athens insomnia scale, the severity of sleep disturbances turned out to be significantly lower in the pre-pandemic period compared to the pandemic period. The same applies to symptoms of depression, as measured by the BDI scale. Changes in depressive symptoms (BDI), insomnia (AIS), and social support between the pre-pandemic and pandemic periods are presented in Table 2.

Next, the same analysis was performed concerning the gender factor (M—male gender; F—female gender). Similar differences were obtained in terms of MOS-SSS. It is important that, in the case of AIS, the differences concern only women. The mean of this variable for the period before the pandemic proved to be statistically significantly lower compared to the pandemic period. The most significant differences were observed in BDI, namely, the score obtained in the current period is statistically significantly higher compared to the time before the pandemic. This applies to both women and men. The results of that analysis are presented in Table 3.

The next analyzed factor was the effect of employment during the pandemic. Statistically, a significant relationship was observed in terms of BDI. The greater difference in scores concerns HCPs who had less work due to the pandemic (Table 4).

Alcohol consumption among HCPs was further evaluated. We analyzed two-week time periods, and asked participants to report an average amount of alcohol consumed daily during every period. The first time period was 2 weeks before the pandemic started, the second one was the first two weeks of the epidemic, the third period was the next 2 weeks, and so forth. The analysis with Friedman’s test showed statistically significant differences between the five time periods in terms of the consumed alcohol dose (λ^2^ (4) = 20.56; *p* < 0.001). The quantity of alcohol consumed in the fifth period proved to be statistically significantly higher compared to the first period (*p* = 0.02). The same applies to the comparisons of periods five with two (*p* = 0.002), four with two (*p* = 0.002) and three with two (*p* = 0.001). The correlation analysis was used to check whether there is a statistically significant relationship between the amount of alcohol consumed and time; as time passed, the HCPs consumed more alcohol (r = 0.08; *p* = 0.049). The increase in alcohol consumption was found to be a harmful result of the pandemic. A further analysis of the factors linked with this phenomenon was performed. The correlations of gender, age, education, living alone vs. with someone, employment status, income, and individuals’ and their relatives’ COVID-19 diagnosis with increased alcohol intake were analyzed.

The statistically significant link with increased alcohol intake concerns those with increased income due to the pandemic (r = 0.46; *p* = 0.03). In the case of HCPs with unchanged income, this relationship also exists, but is weaker (r = 0.09; *p* = 0.09). A statistically significant relationship was also observed in the group who worked more during the pandemic. Those who worked more once the pandemic started consumed even more alcohol over time (r = 0.12; *p* = 0.006).

No correlation between depressive symptoms, sleep disorders or social support and increased alcohol consumption was found. Higher alcohol consumption over time was not linked with more severe depressive or insomnia symptoms. The only factors linked to higher alcohol intake were more work and higher income during the pandemic.

As the statistically significant differences/dependencies mainly concern the level of BDI, a logistical regression analysis was used to check if certain factors affect the increased risk of higher BDI, and if so, which and to what extent.

In the pandemic period, the risk of occurrence of a higher level of BDI concerns women (2.78 times higher risk) (Table 5).

Using logistic regression analysis, it was assessed whether certain factors contributed to the increased risk of depression, and if so, which factors and to what extent. The analyzed variables were dichotomized; the dependent variable was the BDI, for which only gender (an independent variable) was a statistically significant predictor. The other analyzed predictors (independent variables) in this group were not statistically significant. The test values for statistically insignificant predictors were as follows: Income reduction—OR: 1.06; 95% CI: 0.65–1.73; *p* = 0.8. Employment reduction—OR: 0.75; 95% CI: 0.37–1.52; *p* = 0.43. Having money for needs—OR: 0.9; 95% CI: 0.41–2; *p* = 0.8. Increase in alcohol consumption—OR: 1.08; 95% CI: 0.91–1.28; *p* = 0.37.

In other words, the above-mentioned variables simultaneously entered into the model turned out not to be statistically significant predictors of depression symptoms (in the presence of a statistically significant gender predictor).

Some of the variables were not taken into account in the logistic regression due to the low number of cases in the subgroups, e.g., currently performing work, COVID-19 diagnosis, etc. For example, for those doing over 90% of their work in the workplace, the diagnosis of COVID-19 concerned only a few individuals, most people had higher education, and a statistically significantly small number of people lived alone.

Among the analyzed factors of gender, age, education, living (alone vs. with someone), work situation, income, and an individual’s and their relatives’ COVID-19 diagnosis, no other significant correlations were revealed when it comes to insomnia and depression.

## 5. Discussion

The research was conducted to explore depressive and insomnia symptoms, as well as alcohol consumption, as the effects of stress caused by the pandemic. Additionally, the level of social support was assessed. Our research suggests that healthcare professionals can be influenced by the COVID-19 pandemic. An increase in depressive and insomnia symptoms as well as higher alcohol consumption was observed in that group, likely as the effect of an epidemic. Some previous publications considered those features among HCPs; however, studies to date have analyzed the prevalence of mental symptoms, rather than the deterioration of mental state. Nevertheless, when compared with other research, our outcomes stay in line with European and world data, wherein a high prevalence of insomnia and depression among HCPs was revealed during the pandemic. Additionally, the female gender has been reported as the most commonly named factor correlated with a higher prevalence of anxiety, insomnia and depression [35].

It is noteworthy that social support as measured by the MOS-SSS increased significantly in our group of HCPs, which is the only positive information coming from the analysis. In particular, it was assessed that social support can play an important role when previous data from other countries are taken into consideration. As reported by Chinese and Israeli authors, social support and supportive relationship play the role of important protective factors against a worsening of mental state in HCPs during the COVID-19 pandemic [21,22,23]. However, there was no significant correlation observed between social support and depressive symptoms in the current study.

In our study, income and employment were important factors linked with depression and alcohol consumption. Such an observation may be valuable in these times of a global economic crisis developing as a result of the COVID-19 pandemic. Although the effects of the economic crisis were not visible yet during the study, they should be taken into account in the near future. The analyses show that the problems of employment and income may moderate mental state. However, among healthcare professionals, work overload causes stress and tiredness. Our analysis stays in line with previous studies. As noted in another study, a longer working time, which leads to fatigue and tiredness, can be linked with a higher prevalence of depression and anxiety [17]. On the other hand, the fear of job loss and a lower income seem to be important factors in developing depression. During the COVID-19 pandemic, many people lost their jobs, and it is estimated that even more have descended below the poverty line [36]. According to the presented outcomes, maintaining employment and financial support for people needing these may have an influence on lowering the risk of depression, which can be secondary to the COVID-19 crisis. It should be mentioned that in Poland, healthcare professionals are commonly underpaid, with no significant financial resources and savings. Fear of employment loss is one of the most common problems around the world now, but the examined population is different and in an opposite condition. First-line HCPs are considered traumatized in pandemic situations by some authors examining their mental condition [37], and, similarly to in our outcomes, work plays an important role here. However, while increased weekly workload was linked with stress, anxiety and depression in the study of Elbay et al. [38], in our research, less working hours was linked with more severe depressive symptoms (opposite to alcohol intake). One of the possible explanations for this is basic financial status. If it is unstable and insufficient, and if working hours are being limited, then people may be afraid of their future economic situation.

Another problem widely observed and mentioned in the literature is alcohol misuse due to the stress, anxiety, insomnia and depression as a self-medication [28]. However, in the presented study, no correlation between alcohol consumption and depression or insomnia was observed.

In our population of HCPs, alcohol consumption increased in correlation with increased work time and higher income. None of the studies conducted so far indicated that such factors correlated with higher alcohol consumption among HCPs. The correlations that have been identified with alcohol consumption so far are low social support and stigma [39]. Alcohol seems to be a relevant issue for HCPs during pandemics, and the numbers are alarming. In an American multicenter study, among 1132 HCPs who completed the study survey, 42.6% had probable alcohol use disorders [40]. In the same study, as many as 14% had probable major depression [40]. Further studies are needed to indicate risk factors of depression, insomnia and alcohol misuse among healthcare professionals working on the frontline during the COVID-19 pandemic in order to provide appropriate help and counseling for those at risk. Social support appears to be an important protective factor for the mental health of HCPs in the light of previous studies.

## 6. Limitations of the Study

The study has some important limitations that need to be taken into consideration during the interpretation of our outcomes. The first limitation of the study is that the gathered data were self-reported. The authors also acknowledge that the survey assessing mental state in the past is prone to being biased. However, we have assumed that the subjective impression of alteration in an individual’s mental state is crucial. After all, the psychiatric medical interview is based on the patient’s feeling that their current mental state is worse than it was before. As psychiatrists and psychotherapists, we make daily therapeutic decisions based on interviews. All subjects filled out the questionnaire by answering the same question twice (one by one), according to their current and their pre-pandemic state, to allow the direct comparison of each symptom. Psychiatrists in their daily practice evaluate symptoms in accordance with patients’ reports, and also based on the BDI and AIS. There are no fully objective measures for evaluating depressive symptoms free of subjectivity. The patient’s subjective impression of deteriorations or improvements in their mental state remains the most important outcome, and further, the cause of the therapeutic intervention.

Another limitation is that participants were asked about symptoms they had developed several weeks before the study, which could be associated with the inaccurate reproduction of information. Despite these concerns, however, none of the respondents reported problems with remembering their well-being in the period about which they were asked, and none of them reported problems with that issue.

In addition, the cross-sectional design of the study, which was based on two evaluations carried out at the same time point, could be a source of bias. Such an approach is clearly less relevant than the assessment that would be performed at two different time points. The longitudinal design of the study would be more appropriate and precise, but unfortunately, the course of pandemic did not allow that. Thus, many sources of bias might have interfered with the pre-epidemic evaluation of the mental state of the subjects. For example, problems remembering, the impact of current stress, and fear for the future should all be taken under consideration while interpreting the outcomes. Usually, for cross-sectional studies, when the outcome and exposure are measured at the same time, establishing causal relationships is relatively difficult.

Thus, it must be underlined that, given the cross-sectional and retrospective nature of the current study, the outcomes should be interpreted with caution.

## 7. Conclusions

This study on the pandemic’s influence on healthcare professionals’ mental state might be of scientific and clinical significance. We found that:Depressive symptoms increased significantly during the COVID-19 pandemic among the examined healthcare professionals.
When analyzed by gender a significant increase was observed among both women and men;Female gender was, however, linked with an almost three times higher risk of increased depressive symptoms;Those who worked less during the pandemic had more severe depressive symptoms;Insomnia symptoms increased significantly during the COVID-19 pandemic among healthcare professionals.
When analyzed by gender, the significant difference was observed only among women;
Alcohol consumption increased significantly during the COVID-19 pandemic among healthcare professionals. As time passed, the HCPs consumed more alcohol.
Higher alcohol consumption was linked with higher workload and higher income during the pandemic;Social support level increased significantly during the COVID-19 pandemic among healthcare professionals.

Our results suggest that the HCPs involved in the fight against COVID-19 need support and special attention due to the excessive stress load during the pandemic. We assume that the study outcome will help to elucidate to healthcare workers that their mental well-being may be endangered, and that they may need psychological intervention at some point. Mental healthcare professionals should be aware of and sensitive to the symptoms of depressive and sleeping disorders that may appear among healthcare professionals in the face of pandemics, as well as the signals of alcohol misuse. Understanding the background of gender differences may lead to new opportunities for individualized therapy and prevention. The presented outcomes may also draw the medical staff’s attention to their mental condition. The early diagnosis and treatment of mental disorders continue to be of paramount importance in treatment efficacy.

## Figures and Tables

**Table 1 jcm-10-01280-t001:** Demographics.

Variable		*n*	%
Marital status	Bachelor/Miss	34	21.5
	Divorced	2	1.3
	In separation	3	1.9
	In an informal relationship	30	19
	Widower/Widow	0	0
	Married	89	56.3
Gender	Women	92	58.2
	Men	66	41.8
Living alone		34	21.5

**Table 2 jcm-10-01280-t002:** Comparison of social support, sleep disorders and depressive symptoms in the pre-pandemic period and during the pandemic among healthcare professionals.

Variable		M	Me	SD	Min	Max
Social support: MOS-SSS	Pre-pandemic	76.73	81	17.71	28	95
	During pandemic	83.92	88	18.97	30	190
MOS-SSS Statistical test result *		Z = 9.5; *p* < 0.001				
Insomnia: AIS	Pre-pandemic	3.08	3	2.22	0	13
	During pandemic	3.78	3.5	2.52	0	12
AIS Statistical test result *		Z = 3.9; *p* < 0.001				
Depression: BDI	Pre-pandemic	3.86	1	6.98	0	52
	During pandemic	8.49	5.5	9.13	0	48
BDI Statistical test result *		Z = 7.92; *p* < 0.001				

MOS-SSS—Social Support Scale; AIS—Athens Insomnia Scale; BDI—Beck Depression Inventory; * Wilcoxon test.

**Table 3 jcm-10-01280-t003:** Comparison of social support, sleep disorders and depressive symptoms in the pre-pandemic period and during the pandemic among healthcare professionals depending on gender.

Variable		M		Me		SD		Min		Max	
		F	M	F	M	F	M	F	M	F	M
Social support: MOS-SSS	Pre-pandemic period	75.08	79.04	79.5	85	18.23	16.82	29	28	95	95
	During pandemic	82.71	85.62	87	90	20.07	17.33	33	30	190	110
Statistical test result *		F	Z = 7.34; *p* < 0.001								
		M	Z = 6.1; *p* < 0.001								
Insomnia: AIS	Pre-pandemic period	3.22	2.86	3	2	2.09	2.41	0	0	9	13
	During pandemic	4.24	3.15	4	3	2.47	2.46	0	0	12	12
Statistical test result *		F	Z = 3.98; *p* < 0.001								
		M	Z = 1.15; *p* = 0.25								
Depression: BDI	Pre-pandemic period	5.02	2.24	1	1	8.51	3.46	0	0	52	16
	During pandemic	10.75	5.35	9	3	10.13	6.36	0	0	48	31
Statistical test result *		F	Z = 6.59; *p* < 0.001								
		M	Z = 4.18; *p* < 0.001								

MOS-SSS—Social Support Scale; AIS—Athens Insomnia Scale; BDI—Beck Depression Inventory; * Wilcoxon test, F—female gender; M—male gender.

**Table 4 jcm-10-01280-t004:** The effect of the pandemic on the employment of healthcare professionals (HCPs) and the increase in their Beck depression inventory (BDI) scores between the two time periods.

	Effect of the Pandemic on Employment	M	Me	SD	Min	Max	Statistical Test Result *
BDI	Unchanged	2.4	2	6.8	−34	14	U = 1666; *p* = 0.02
	Less work	5.57	3	7.65	−10	39	

BDI—Beck Depression Inventory; * Mann–Whitney U test.

**Table 5 jcm-10-01280-t005:** Gender is a statistically significant predictor in the studied group of healthcare professionals influencing the risk of a high BDI score.

Variable	Odds Ratio *	95% CI
Gender (female)	2.78	1.05–7.36 *p* = 0.04

BDI—Beck Depression Inventory; * Logistic regression.

## Data Availability

Please contact the corresponding author with any data availability requests.

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
