# Peer review of "COVID-19 Pandemic Influence on Healthcare Professionals"

_jcm, 2021, doi:10.3390/jcm10061280_

Round 1

Reviewer 1 Report

jcm-1105556

I would like to thank the authors for contributing their work on such an important and timely topic. This study uses a retrospective cross-sectional design to examine how healthcare professionals have been impacted by COVID-19. Overall, this study has several important takeaways. Below are some critiques that may help improve the manuscript:

Introduction:

  1. The authors build a set of arguments of how the stress of COVID-19 may impact healthcare professionals. There is quite a bit related to neuro-circuits and that is not tested here. I would encourage the authors to look more broadly at stress that involves outcomes tested in the current paper.
  2. The introduction makes a bit of a leap from the COVID-19 pandemic. This type of stressor may be different than other stressors that the authors reference. I would make sure to include literature from the pandemic that has been tested in other populations like this during COVID in order to draw on the parallels.
  3. The authors need to include additional literature on the gender differences. Are you saying there are gender differences with others stressors or with COVID-19? I can think of several reasons why COVID-19 impacts women at higher rates (and some have been published) but it is important to clarify that.
  4. The background literature does not quite set up the questions asked in the objective section. It seems a bit vague to me and I would encourage the authors to streamline this.
  5. The authors discuss the impact on cognitive functioning but it does not seem to be tested directly in the current study.
  6.  

Method/Results:

  1. Did the authors collect any data on healthcare workers that do not work with covid-19 patients? This would be an interesting comparison group (or time spent with covid-19 patients).
  2. Although not the perfect measures of suicidality, did the authors consider looking at item 9 of the BDI-II to examine covid-19 on suicidality?
  3. Can the authors include cronbach’s alpha for the scales used?
  4. The authors use gender for most of the paper and then switch to sex. Please clarify which construct was measured.
  5. It looks as if the MOS-SSS went up during the pandemic overall. This is something that should be explore more later.
  6. It might be helpful to have a correlation table with the variables and the significant correlations highlighted.

Discussion:

  1. Given the cross sectional and retrospective nature of the current study, it may be in the best interest of the paper to make less strong statements about the conclusions drawn from these data.
  2. There needs to be further discussion of the potential impact of the results found. Similarly, the authors do not discuss the MOS-SSS results significantly.
  3. The authors also do not spend enough time discussing limitation in detail. The retrospective and cross-sectional design have severe limitations.

Minor Notes:

  1. Do the authors want to use the term medics and healthcare providers interchangeably? I think it is actually important to define who was included as a healthcare provider.
  2. Line 40 contains a typo of “menntal”
  3. Typo on line 84 “t”

Author Response

Reviewer 1

Thank you for your valuable review. We deeply appreciate your evaluation.  We found all comments very important and we made our best to improve the manuscript. We went thru all comments and below we are providing point-by-point answers to each criticism.

All changes can be seen in the manuscript as a “track changes” option. Please find each comment addressed below:

Introduction:

  1. The authors build a set of arguments of how the stress of COVID-19 may impact healthcare professionals. There is quite a bit related to neuro-circuits and that is not tested here. I would encourage the authors to look more broadly at stress that involves outcomes tested in the current paper.

We have deleted irrelevant and unnecessary sentences form the introduction section, adding more appropriate information (cited below).

  1. The introduction makes a bit of a leap from the COVID-19 pandemic. This type of stressor may be different than other stressors that the authors reference. I would make sure to include literature from the pandemic that has been tested in other populations like this during COVID in order to draw on the parallels.

We have significantly changed our approach to the introduction to the issue discussed. Instead of  neurochemical problems, we have described the research to date on the impact of the pandemic on the mental state of HCPs.

Now we are writing:

“In the time of an epidemic, HCPs experience many stressors that affect well-being. First group of stressors are those directly related to the nature of their job. Second group of stressors are those applying to every human being in the general population. The fear of being infected, uncertainty of the future, fear of deterioration of the financial situation as well as illness and loss of loved ones concern both HCPs and the general population. Both kinds of stressful stimulus have an essential impact on HCPs’ functioning and their mental condition. Importantly, the wider knowledge about the SARS-CoV-2, COVID-19 and its complications the more fear for their and loved ones' health in this group.

In a systematic review on mental health of healthcare personnel during COVID-19 in Asia the authors mentioned depression, insomnia and anxiety as the most common psychiatric effects of COVID-19 pandemic [8]. There are some important reports from European countries as well, e.g. Italy [9], Spain [10), Norway [11] or Switzerland [12]

 According to previous research there are several most important risk factors of HCP s’ mental health deterioration during pandemic. Those that were identified so far are: female gender [13-15], direct contact with the infected patients [16, 17], especially with the lack of appropriate personal protective equipment [18]. Also shorter work experience predicted higher prevalence of anxiety and depressive symptoms [19]. Healthcare professionals  both significantly younger or older than average, were also at risk of mental disorders due to the pandemic [20-22].

Some researches also indicated protective factors for pandemic induced mental state worsening. First of those is mental resilience [23], the second social and family support [24,25]. “

  1. The authors need to include additional literature on the gender differences. Are you saying there are gender differences with others stressors or with COVID-19? I can think of several reasons why COVID-19 impacts women at higher rates (and some have been published) but it is important to clarify that.

Originally we meant female gender in the context of other, general stressors. We admit that our statement on that topic was confusing and it was already corrected. Now we are also additionally citing the literature that argues for women to be more sensitive to the pandemic stress. It can be found in comprehensively reformulated introduction section.

  1. The background literature does not quite set up the questions asked in the objective section. It seems a bit vague to me and I would encourage the authors to streamline this.

As we mentioned above we reached and have analyzed the publications linked directly to the objective section.

  1. The authors discuss the impact on cognitive functioning but it does not seem to be tested directly in the current study.

That part of introduction was now deleted as suggested.

Method/Results:

  1. Did the authors collect any data on healthcare workers that do not work with covid-19 patients? This would be an interesting comparison group (or time spent with covid-19 patients).

Unfortunately we did not gather the data form other (“non-COVID”)  HCPs, however it would be definitely interesting to compare such populations. In fact presented analysis is a part of a broader study where control group and psychiatric patients were examined as well. We will be comparing those groups subsequently in the future.

  1. Although not the perfect measures of suicidality, did the authors consider looking at item 9 of the BDI-II to examine covid-19 on suicidality?

Thank you for that suggestion. We will examine the suicidality in another part of the analysis.

  1. Can the authors include cronbach’s alpha for the scales used?

We now underlined in the text that Cronbach’s alpha for all measures was higher than 0.7 stating that: “Cronbach’s alpha for all measures was higher than 0.7 therefore, this study can be considered reliable.”

  1. The authors use gender for most of the paper and then switch to sex. Please clarify which construct was measured.

We should have used “gender” as the participants were asked the question in the simplest way: “what is your gender?”. The study was conducted on Polish population and what is important, in Polish language there is no distinction between those two constructs “sex” and “gender”. Only one word is describing that matter: “płeć”. To avoid misleading wording we now use only “gender” thru all the manuscript.

  1. It looks as if the MOS-SSS went up during the pandemic overall. This is something that should be explore more later.

We will be happy to work on that in the next step of the analysis.

  1. It might be helpful to have a correlation table with the variables and the significant correlations highlighted.

Discussion:

  1. Given the cross sectional and retrospective nature of the current study, it may be in the best interest of the paper to make less strong statements about the conclusions drawn from these data.

We have reformulated our conclusions and discussion as suggested, avoiding such strong and categorical statements

  1. There needs to be further discussion of the potential impact of the results found. Similarly, the authors do not discuss the MOS-SSS results significantly.

We have added some discussion about social support. Now we are writing that:

Noteworthy the social support as measured by MOS-SSS increased significantly in case of analyzed group and that can play an important role when previous data from other countries is taken under consideration. As reported by Chinese and Israel authors the social support and supportive relationship play the role of important protective factors against mental state worsening in HCPs during COVID-19 pandemic [23-25].”

  1. The authors also do not spend enough time discussing limitation in detail. The retrospective and cross-sectional design have severe limitations.

We prepared separate section of the limitations and completed it with additional information accordingly to the suggestion. We are now stating that:

Also a cross-sectional design of the study, which was based on two evaluations carried out at the same time point can be a subject of bias. Such approach is clearly less relevant than the assessment that would be performed at two different time points. Longitudinal design of the study would be more appropriate and precise, but unfortunately the course of pandemic did not let for that. Thus many sources of bias might have interfered with the pre-epidemic evaluation of mental state of the subjects For instance: problems remembering, the impact of current stress, fear for the future, should be taken under consideration while interpreting the outcomes.  Usually for cross-sectional studies, when the outcome and exposure were measured at the same time, establishing causal relationships is relatively difficult.

Minor Notes:

  1. Do the authors want to use the term medics and healthcare providers interchangeably? I think it is actually important to define who was included as a healthcare provider.

We unified naming into HCP – healthcare professionals, and defined in more details the analyzed group, as well as desribed the recruitment process.

  1. Line 40 contains a typo of “menntal”

We’ve made corrections as suggested.

  1. Typo on line 84 “t”

W made corrections as suggested.

Thank you for such detailed comments which let us to improve the manuscript. We hope that all those changes will meet your acceptation. We addressed all the comments as suggested above.

Reviewer 2 Report

General comment

The present manuscript tried to examine alcohol intake, sleep disorders, and depressive symptoms during the COVID-19 pandemic in comparison with the pre-pandemic period in a group of healthcare workers (HCWs), using a cross-sectional design. The possible associations between sociodemographic variables and those psychological factors were also investigated.

The paper deals with a clinically interesting topic, given the negative effects that the COVID-19 outbreak has been shown to have especially among healthcare professionals. However, it presents some important methodological weak points that need to be addressed.

Introduction

In the introduction section, the authors described the main topics investigated in their study. However, in my opinion, some improvements could be made. First, I would suggest the authors to include additional references that examined the prevalence of mental health symptoms among healthcare workers also in Western countries (e.g., Di Tella, M., Romeo, A., Benfante, A., & Castelli, L. (2020). Mental health of healthcare workers during the COVID‐19 pandemic in Italy. Journal of evaluation in clinical practice, 26(6), 1583-1587; Gómez-Salgado, J., Domínguez-Salas, S., Romero-Martín, M., Ortega-Moreno, M., García-Iglesias, J. J., & Ruiz-Frutos, C. (2020). Sense of coherence and psychological distress among healthcare workers during the COVID-19 pandemic in Spain. Sustainability, 12(17), 6855).

Secondly, I would suggest the authors to reformulate the second part of the introduction, as a clear link between the different aspects described seems sometimes to be missing (e.g., in the description of the association between mental health symptoms and alcohol consumption). In addition, the authors largely described the neural basis of chronic stress; however, in their study, they did not take into account this level of examination.  Finally, the following sentences “Gender factor was also evaluated in the conducted study. Understanding the background of these differences may lead to new opportunities for individualized therapy and prevention” seem to be part of the objective and conclusion sections, respectively.

In the “Objectives” paragraph, I would ask the authors to define clearly that the pre- and post- COVID-19 outbreak assessments have been conducted simultaneously and not by using a longitudinal evaluation of mental health symptoms. Also, I would suggest the authors to report first the main aims of the study and then the related hypotheses for each of those goals.

Methods

I would suggest the authors to split this section into different subsections (e.g., Participants, Procedure, Measures, and Statistical analysis), in order to make the description of the different aspects clearer. Regarding the participants, I would ask the authors: (1) to indicate the number of HCWs that were initially contacted to take part in the study, (2) to include information about the inclusion/exclusion criteria that have been used for the recruitment of participants, and (3) to indicate how the HCWs have been contacted to participate in the study.

In the measure description, I would ask the authors to include the references for the Polish version of the questionnaires they administered and to add information about the psychometric properties of those instruments.

With regard to the statistical analyses the authors performed, I would ask them to indicate whether and how normality of distribution has been verified before carrying out the analysis. In addition, more information about the data analysis planning should be provided, in order to make clear how the different aims of the study have been tested. Finally, I would ask the authors to explain why they did not perform more complex data analysis, such as multiple regressions, to analyse the associations between their target variables.

Results

With regard to this section, I have the following concerns:

  • I guess that an error might have been occurred in the uploading of the tables, as some information seems to be missing (e.g., Tables 1 and 3). In addition, a legend is not present for each of the tables that have been reported.
  • I would ask the authors to explain why more appropriate tests, such as a mixed ANOVA, have not been used for the analysis of gender x time differences.
  • It is not clear how the effect of employment has been evaluated and why it has been considered only in regard to depression scores.
  • The associations between alcohol consumption and the other variables assessed seem to be described in a quite confusing way. I would suggest the authors to explain clearly the different relationships that have been evaluated and the reason behind those analyses.
  • Finally, the logistic regression analysis that has been mentioned in Table 5 comes out of nowhere. Indeed, no mention has been made to this analysis in the description of data analysis planning and it is not clear why and how it has been performed.

Discussion

In this final section, the authors discussed the main results of their study. However, I think that some aspects could be examined better. Particularly, I would suggest the authors to link better the main findings of the study with each other and to consider also the available literature on the topic in the discussion of those results. Moreover, the description of the main results of correlation analyses seems to be quite confusing and could thus be rephrased, in order to make the presentation of those findings clearer to the reader. Also, I would suggest the authors to present the limitations of the study as a separate subsection of the discussion. Finally, I agree with the considerations the authors made about the subjectivity intrinsic to the psychological symptoms the individuals referred. However, a cross-sectional design with two evaluations carried out at the same time is very different from an assessment that is performed at two different time points (i.e. using a longitudinal design), as many sources of bias might have interfered with the pre-epidemic evaluation and this should be clearly stated in the limitation section.

Minor corrections

Please, correct some typos throughout the text and use the acronym HCPs homogenously in the manuscript.

Author Response

Reviewer 2

Thank you for your valuable review. We deeply appreciate your evaluation.  We found all comments very important and we made our best to improve the manuscript. We went thru all comments and below we are providing point-by-point answers to each criticism.

All changes can be seen in the manuscript as a “track changes” option. Please find each comment addressed below:

General comment

The present manuscript tried to examine alcohol intake, sleep disorders, and depressive symptoms during the COVID-19 pandemic in comparison with the pre-pandemic period in a group of healthcare workers (HCWs), using a cross-sectional design. The possible associations between sociodemographic variables and those psychological factors were also investigated.

The paper deals with a clinically interesting topic, given the negative effects that the COVID-19 outbreak has been shown to have especially among healthcare professionals. However, it presents some important methodological weak points that need to be addressed.

Introduction

In the introduction section, the authors described the main topics investigated in their study. However, in my opinion, some improvements could be made. First, I would suggest the authors to include additional references that examined the prevalence of mental health symptoms among healthcare workers also in Western countries (e.g., Di Tella, M., Romeo, A., Benfante, A., & Castelli, L. (2020). Mental health of healthcare workers during the COVID‐19 pandemic in Italy. Journal of evaluation in clinical practice, 26(6), 1583-1587; Gómez-Salgado, J., Domínguez-Salas, S., Romero-Martín, M., Ortega-Moreno, M., García-Iglesias, J. J., & Ruiz-Frutos, C. (2020). Sense of coherence and psychological distress among healthcare workers during the COVID-19 pandemic in Spain. Sustainability, 12(17), 6855).

We are now providing additional references as suggested. We mentioned studies proposed by Reviewer, as well as few others corresponding with European studies.

Secondly, I would suggest the authors to reformulate the second part of the introduction, as a clear link between the different aspects described seems sometimes to be missing (e.g., in the description of the association between mental health symptoms and alcohol consumption).

We have strongly reformulated the introduction section, and also added the information about the linkage between AUD and mental health symptoms, stating that:

“The coexistence of alcohol abuse and dependence with some mental disorders, especially mood and anxiety, has already been studied and fully described [30]. Higher rates of both anxiety and depression have been proven among alcohol abusers [31]. This link seems to be particularly important in the context of chronic stress in the face of a pandemic”

In addition, the authors largely described the neural basis of chronic stress; however, in their study, they did not take into account this level of examination. 

We have significantly changed our approach to the introduction to the issue under discussion. Instead of neurochemical problems, we have described the research to date on the impact of the pandemic on the mental state of HCPs.

Now we are writing:

In the time of an epidemic, HCPs experience many stressors that affect well-being. First group of stressors are those directly related to the nature of their job. Second group of stressors are those applying to every human being in the general population. The fear of being infected, uncertainty of the future, fear of deterioration of the financial situation as well as illness and loss of loved ones concern both HCPs and the general population. Both kinds of stressful stimulus have an essential impact on HCPs’ functioning and their mental condition. Importantly, the wider knowledge about the SARS-CoV-2, COVID-19 and its complications the more fear for their and loved ones' health in this group.

In a systematic review on mental health of healthcare personnel during COVID-19 in Asia the authors mentioned depression, insomnia and anxiety as the most common psychiatric effects of COVID-19 pandemic [8]. There are some important reports from European countries as well, e.g. Italy [9], Spain [10), Norway [11] or Switzerland [12]

 According to previous research there are several most important risk factors of HCP s’ mental health deterioration during pandemic. Those that were identified so far are: female gender [13-15], direct contact with the infected patients [16, 17], especially with the lack of appropriate personal protective equipment [18]. Also shorter work experience predicted higher prevalence of anxiety and depressive symptoms [19]. Healthcare professionals  both significantly younger or older than average, were also at risk of mental disorders due to the pandemic [20-22].

Some researches also indicated protective factors for pandemic induced mental state worsening. First of those is mental resilience [23], the second social and family support [24,25]

 Finally, the following sentences “Gender factor was also evaluated in the conducted study. Understanding the background of these differences may lead to new opportunities for individualized therapy and prevention” seem to be part of the objective and conclusion sections, respectively.

Both sentences were removed from the introduction section. Second one was moved into conclusions. Gender factor had already been mentioned in objectives thus we did not repeat that information, and simply removed that.

In the “Objectives” paragraph, I would ask the authors to define clearly that the pre- and post- COVID-19 outbreak assessments have been conducted simultaneously and not by using a longitudinal evaluation of mental health symptoms.

We agree that there was no directly clear statement on that issue in the manuscript. Thank you for that suggestion. We completed the objective section as suggested by stating:

“Pre-pandemic and post-pandemic outbreak assessments have been conducted simultaneously. Longitudinal evaluation of mental health symptoms was not performed as it was impossible due to unexpected nature of COVID-19 epidemic.”

Also, I would suggest the authors to report first the main aims of the study and then the related hypotheses for each of those goals.

We have rearranged the objective section to be more clear and orderly, starting with aims then ending with hypotheses.

Methods

I would suggest the authors to split this section into different subsections (e.g., Participants, Procedure, Measures, and Statistical analysis), in order to make the description of the different aspects clearer.

We divided Methods section as suggested by Reviewer.

Regarding the participants, I would ask the authors: (1) to indicate the number of HCWs that were initially contacted to take part in the study, (2) to include information about the inclusion/exclusion criteria that have been used for the recruitment of participants, and (3) to indicate how the HCWs have been contacted to participate in the study.

We are now providing additional information, explaining more about the recruitment process. We are writing that:

“HCPs group consisted of medical doctors, paramedics, and nurses, who were recruited using the snowball method. In total 186 HCPs were invited to participate. The refusal rate was as high as 15%. At the beginning 20 HCPs from one of the Warsaw Covid-19 units were asked to participate. Those who took part in the study provided referrals to recruit another HCP working on COVID-19 unit. Every HCP willing to participate, who met criteria of working in pandemic frontline met eligibility criteria and was included to the analysis.”

In the measure description, I would ask the authors to include the references for the Polish version of the questionnaires they administered and to add information about the psychometric properties of those instruments.

We added required information as suggested.

With regard to the statistical analyses the authors performed, I would ask them to indicate whether and how normality of distribution has been verified before carrying out the analysis. In addition, more information about the data analysis planning should be provided, in order to make clear how the different aims of the study have been tested. Finally, I would ask the authors to explain why they did not perform more complex data analysis, such as multiple regressions, to analyse the associations between their target variables.

We are now providing some more information about statistical analysis as suggester by Reviewer stating that:

“Statistical analysis: Statistical analysis was conducted using the IBM SPSS Statistics 25 Package. At the beginning insomnia, depression and alcohol consumption were compared between two analyzed time periods. Further the correlation of factors such as: gender, age, education, living alone vs. with someone, employment status, income, individuals ’ and their relatives’ COVID-19 diagnosis with insomnia, depression and alcohol consumption increase was. The Wilcoxon test was used to check whether there are statistically significant differences between the two time periods for the analyzed variables. The analysis of the Spearman correlation allowed to check the presence of statistically significant relationship between the studied variables. Then a multinomial logistic regression was performed for factors significantly correlated with the increase in depression and insomnia symptoms and higher alcohol consumption. Significant outcomes of logistic regression are presented in the result section.

Non-parametric tests were used. due to non normal distribution of variables as well as the groups had different number of participants. The logistic regression analysis did not reveal any statistically significant predictors other than those described in the results, i.e. those that had an impact on the other scores of the scales used in survey.”

  1. Results:

 With regard to this section, I have the following concerns:

  • I guess that an error might have been occurred in the uploading of the tables, as some information seems to be missing (e.g., Tables 1 and 3). In addition, a legend is not present for each of the tables that have been reported.

We completed missing legends, and ordered table 1.

  • I would ask the authors to explain why more appropriate tests, such as a mixed ANOVA, have not been used for the analysis of gender x time differences.

We did not use ANOVA because the compared groups were unequal or small. Mainly for this reason, non-parametric tests were used. It was also due to non normal distribution of variables. Such explanation can be found now in “statistical analysis” section as requested above.

  • It is not clear how the effect of employment has been evaluated and why it has been considered only in regard to depression scores.

Participants were asked “ Do you spend more time at work on duty since the pandemic had started?”

We analyzed alcohol consumption and insomnia in regard to as well. It is stated in the manuscript that:

“Statistically significant link with increased alcohol intake concerns those with income increased due to the pandemic, r = 0.46; p = 0.03. In the case of HCPs with income unchanged, this relationship also exists, but is weaker, r = 0.09; p = 0.09. A statistically significant relationship was also observed in the group who worked more during the pandemic. Those who worked more, since pandemic had started consumed alcohol  even more over time r = 0.12; p = 0.006.”

  • The associations between alcohol consumption and the other variables assessed seem to be described in a quite confusing way. I would suggest the authors to explain clearly the different relationships that have been evaluated and the reason behind those analyses.

We are now providing additional information on how alcohol consumption was assessed and measured.  We also presented in more details what relationships were studied. We are now writing that:

“The increase of alcohol consumption was found to be a harmful result of pandemic.. Further analysis of factors linked with that phenomenon was performed. Correlation of gender, age, education, living alone vs. with someone, employment status, income, individuals ’ and their relatives’ COVID-19 diagnosis with increased alcohol intake were analyzed.

Statistically significant link with increased alcohol intake concerns those with income increased due to the pandemic, r = 0.46; p = 0.03. In the case of HCPs with income unchanged, this relationship also exists, but is weaker, r = 0.09; p = 0.09. A statistically significant relationship was also observed in the group who worked more during the pandemic. Those who worked more, since pandemic had started consumed alcohol  even more over time r = 0.12; p = 0.006.No correlation between depressive symptoms, sleep disorders nor social support with increasing alcohol consumption was found. Higher alcohol consumption over time was not linked with more severe depressive nor insomnia symptoms. The only factors linked to higher alcohol intake were more work and higher income during the pandemic”

  • Finally, the logistic regression analysis that has been mentioned in Table 5 comes out of nowhere. Indeed, no mention has been made to this analysis in the description of data analysis planning and it is not clear why and how it has been performed.

We are now providing in the method section more information on how logistic regression was planned and performed. We have explained in method section/statistical analysis that:

“The analysis of the Spearman correlation allowed to check the presence of statistically significant relationship between the studied variables. Then a multinomial logistic regression was performed for factors significantly correlated with the increase in depression and insomnia symptoms and higher alcohol consumption. Significant outcomes of logistic regression are presented in the result section”

Discussion

In this final section, the authors discussed the main results of their study. However, I think that some aspects could be examined better. Particularly, I would suggest the authors to link better the main findings of the study with each other and to consider also the available literature on the topic in the discussion of those results.

We have widened the discussions as recommended and considered more literature in regards to analyzed outcomes.

 Moreover, the description of the main results of correlation analyses seems to be quite confusing and could thus be rephrased, in order to make the presentation of those findings clearer to the reader.

We have rephrased and completed discussion section, hoping that the text is clearer now.

Also, I would suggest the authors to present the limitations of the study as a separate subsection of the discussion. Finally, I agree with the considerations the authors made about the subjectivity intrinsic to the psychological symptoms the individuals referred. However, a cross-sectional design with two evaluations carried out at the same time is very different from an assessment that is performed at two different time points (i.e. using a longitudinal design), as many sources of bias might have interfered with the pre-epidemic evaluation and this should be clearly stated in the limitation section.

We prepared separate section of the limitations and completed it with additional information accordingly to the suggestion.

We are now stating that: ” Also a cross-sectional design of the study, which was based on two evaluations carried out at the same time point can be a subject of bias. Such approach is clearly less relevant than the assessment that would be performed at two different time points. Longitudinal design of the study would be more appropriate and precise, but unfortunately the course of pandemic did not let for that. Thus many sources of bias might have interfered with the pre-epidemic evaluation of mental state of the subjects For instance: problems remembering, the impact of current stress, fear for the future, should be taken under consideration while interpreting the outcomes. Usually, for cross-sectional studies, when the outcome and exposure were measured at the same time, establishing causal relationships is relatively difficult”.

 Minor corrections

Please, correct some typos throughout the text and use the acronym HCPs homogenously in the manuscript.

We went through the entire document and corrected typos. Throughout the manuscript, we also standardized the nomenclature of the analyzed group on HCP.

Thank you for such detailed comments which let us to improve the manuscript. We hope that all those changes will meet your acceptation. We addressed all the comments as suggested.

Round 2

Reviewer 2 Report

I really appreciate the changes that the authors have made to the paper. It has been adjusted and improved, accordingly to the reviewers’ suggestions.

I have just a few more comments for further improving the article that are reported below.

  • In the introductive section, the acronym “HPs” should be added the first time the word “healthcare professionals” has been mentioned in the text (line 44 of the manuscript). Moreover, once it has been mentioned the first time, it can be used homogenously throughout the text.
  • In the description of the study objectives, I would suggest the authors to make the following sentence “It was examined whether the change in the abovementioned factors was associated with the following…” clearer, explaining, for instance, that the “above-mentioned factors” refer to insomnia, alcohol intake, and symptoms of depression. In addition, I think that the following phrase “In particular, while SARS–CoV–2 129 was spreading in Poland, the influence of the situation on the mental state of health care professionals was explored” could be moved at the beginning of this subsection or deleted from the text.
  • With regard to the measure subsection, I appreciate that the authors have included information about the Cronbach’s alpha values and the Polish versions of the instruments in the revised version of their manuscript. However, I would ask the authors to include that information also for the Social support scale.
  • In the statistical analysis subsection, I would suggest the authors to move the information about the data distribution before reporting the tests that they used for data analysis, in order to make this choice clearer. With regard to the logistic regression the authors performed, I am still not clear how this analysis has been performed. Indeed, the authors did not clearly specify the dependent variable(s) that were included into the model(s) and if a different logistic regression has been performed for each of their target variables (i.e., depressive and insomnia symptoms, and alcohol consumption). Furthermore, have those target variables dichotomized based on a cut-off value before performing the logistic regression(s)? Finally, I would suggest the authors to move the following sentence “The logistic regression analysis did not reveal any statistically significant predictors other than those described in the results, i.e. those that had an impact on the other scores of the scales used in survey” to the result section and to report in the table also the data for the non-significant predictors, in order to make the tested model clearer.
  • With regard to the result section, I would ask the authors to include either in the text or in Table 5 also the statistics for the overall model that has been tested (in addition to the results for all the predictors that have been included into the model).

Author Response

Reviewer 2

Thank you for your acceptance of our work. We made further correction as suggested. All changes can be seen in track changes mode. Please find each comment addressed below:

I really appreciate the changes that the authors have made to the paper. It has been adjusted and improved, accordingly to the reviewers’ suggestions.

I have just a few more comments for further improving the article that are reported below.

  • In the introductive section, the acronym “HPs” should be added the first time the word “healthcare professionals” has been mentioned in the text (line 44 of the manuscript). Moreover, once it has been mentioned the first time, it can be used homogenously throughout the text.

We have added acronym as suggested.

  • In the description of the study objectives, I would suggest the authors to make the following sentence “It was examined whether the change in the abovementioned factors was associated with the following…” clearer, explaining, for instance, that the “above-mentioned factors” refer to insomnia, alcohol intake, and symptoms of depression. In addition, I think that the following phrase “In particular, while SARS–CoV–2 129 was spreading in Poland, the influence of the situation on the mental state of health care professionals was explored” could be moved at the beginning of this subsection or deleted from the text.

We completed information in that section as suggested. We also moved mentioned sentence at the beginning of  the subsection.

  • With regard to the measure subsection, I appreciate that the authors have included information about the Cronbach’s alpha values and the Polish versions of the instruments in the revised version of their manuscript. However, I would ask the authors to include that information also for the Social support scale.

We are now providing Cronbach’s alpha value for MOS-SSS, stating that: ”Cronbach’s alpha for the scale was as high as 0.97.”

  • In the statistical analysis subsection, I would suggest the authors to move the information about the data distribution before reporting the tests that they used for data analysis, in order to make this choice clearer.

We moved the information at the beginning of the subsection as suggested.

  • With regard to the logistic regression the authors performed, I am still not clear how this analysis has been performed. Indeed, the authors did not clearly specify the dependent variable(s) that were included into the model(s) and if a different logistic regression has been performed for each of their target variables (i.e., depressive and insomnia symptoms, and alcohol consumption). Furthermore, have those target variables dichotomized based on a cut-off value before performing the logistic regression(s)?

We are now providing more information about modeling, writing that:

“Using logistic regression analysis, it was assessed whether and if so, which factors and to what extent contributed to the increased risk of depression. Analyzed variables were dichotomized; the dependent variable was the BDI, for which only gender (an independent variable) was a statistically significant predictor. The other analyzed predictors (independent variables) in this group were not statistically significant. The test values for statistically insignificant predictors were as follows: income reduction - OR: 1.06; 95% CI: 0.65-1.73; p = 0.8, employment reduction - OR: 0.75; 95% CI: 0.37-1.52; p = 0.43, having money for needs - OR: 0.9; 95% CI: 0.41-2; p = 0.8, increase in alcohol consumption - OR: 1.08; 95% CI: 0.91-1.28; p = 0.37. In other words, the above-mentioned variables simultaneously entered into the model turned out not to be a statistically significant predictor of depression symptoms (in the presence of a statistically significant gender predictor).

Some of the variables were not taken into account in the logistic regression due to the low number of cases in subgroups, e.g. currently performed work, COVID-19 diagnosis, etc. Examples: over 90% work in the workplace, the diagnosis of COVID-19 concerned only few individuals, most people had higher education, statistically significantly fewer people lived alone.”

  •  Finally, I would suggest the authors to move the following sentence “The logistic regression analysis did not reveal any statistically significant predictors other than those described in the results, i.e. those that had an impact on the other scores of the scales used in survey” to the result section and to report in the table also the data for the non-significant predictors, in order to make the tested model clearer.

We removed that sentence and discussed the issue in the result section as advised.

  • With regard to the result section, I would ask the authors to include either in the text or in Table 5 also the statistics for the overall model that has been tested (in addition to the results for all the predictors that have been included into the model).

We completed required information, as mentioned above:

“The other analyzed predictors (independent variables) in this group were not statistically significant. The test values for statistically insignificant predictors were as follows: income reduction - OR: 1.06; 95% CI: 0.65-1.73; p = 0.8, employment reduction - OR: 0.75; 95% CI: 0.37-1.52; p = 0.43, having money for needs - OR: 0.9; 95% CI: 0.41-2; p = 0.8, increase in alcohol consumption - OR: 1.08; 95% CI: 0.91-1.28; p = 0.37”